# DEEP NEURAL MAPS

**Mehran Pesteie, Purang Abolmaesumi & Robert Rohling**
Department of Electrical and Computer Engineering
University of British Columbia
Vancouver, BC, Canada
{mehranp, purang, rohling}@ece.ubc.ca

## ABSTRACT

We introduce a new unsupervised representation learning and visualization method using deep convolutional networks and self organizing maps called Deep Neural Maps (DNM). DNM jointly learns an embedding of the input data and a mapping from the embedding space to a two-dimensional lattice. We compare visualizations of DNM with those of t-SNE and LLE on the MNIST and COIL-20 data sets. Our experiments show that the DNM can learn efficient representations of the input data, which reflects characteristics of each class. This is shown via back-projecting the neurons of the map on the data space.

## 1 INTRODUCTION

High dimensional data are very common across different machine learning domains such as computer vision and genomics. Hence, dimensionality reduction and data visualization are important techniques to represent high-dimensional data with their corresponding mapping in low dimension. Linear methods such as principal component analysis (PCA) (Wold et al. (1987)) are not effective in nonlinear data representation. Therefore in order to represent complex structures of data more efficiently, a large number of nonlinear dimensionality reduction algorithms have been proposed (Lee et al. (2015); Yang & Fan (2014)). Such methods aim to preserve the local structures of data while reducing the dimensions. However, it has been argued that these techniques are not very successful at visualizing high-dimensional data (Maaten & Hinton (2008)). In order to overcome this issue, Maaten & Hinton (2008) proposed the t-SNE algorithm, which learns a projection of the data into a matrix of pair-wise similarities.

Representation learning algorithms, whose goal is to capture the principal factors of the observed data (Bengio et al. (2013)), are a common technique for dimensionality reduction. However, the majority of the representation learning algorithms aim to yield an embedding of the data that are efficient for clustering or classification (Guo et al. (2017); Aytekin et al. (2018)). Hence, to interpret the similarities of the embedding space, a visualization algorithm such as t-SNE, needs to be performed after learning the embedding. In this paper, we introduce a method to learn and optimize an embedding space for visualization, called Deep Neural Maps (DNM). In particular, we utilize the self organizing maps (SOM) model (Kohonen (1998)) in conjunction with deep convolutional auto-encoders. The SOM algorithm preserves the topological structures among data points. Unlike the original t-SNE algorithm, our proposed method does not require re-training and provides a map that can be used to visualize a new data point after training.

## 2 DEEP NEURAL MAPS

Figure 1 shows the block diagram of our proposed model. The DNM model consists of a convolutional auto-encoder and an SOM. In an SOM network, the neurons are located at the nodes of a one or two dimensional lattice. Therefore, it provides a topographic map of the input where the locations of the neurons in the map indicate statistical features of the input patterns. Let $\mathcal{E}_\theta : X \to Z$ be an encoding function with parameters $\theta$, which maps input $X$ to embedding $Z$ and $\mathcal{D}_\phi : Z \to \widehat{X}$ be the corresponding decoding function with parameters $\phi$, which takes $Z$ and reconstructs $\widehat{X}$. Also, let the weight vector of each neuron in the SOM be denoted by $w_j = [w_{j1}, w_{j2}, ..., w_{jm}]$

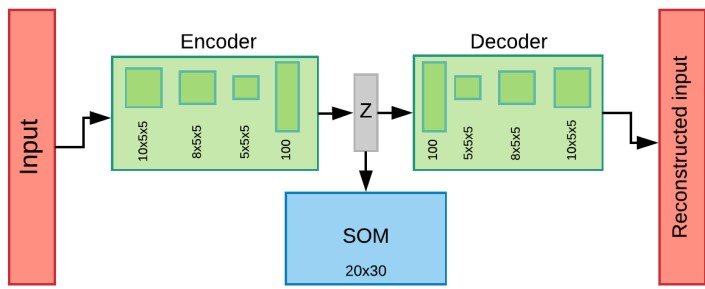

Figure 1: Block diagram of the DNM architecture.

for $j = 1, 2, ..., l$, where $m$ is the dimension of the embedding space and $l$ is the total number of the neurons in the SOM. To find the best matching neuron from $w$ to $Z_i$, one needs to minimize $u = \arg\min_j ||Z_i - w_j||$, $j = 1, 2, .., l$, which is the Euclidean distance between embedding $Z_i$ and weight vector $w_j$. In the DNM model, the objective is to jointly optimize $\theta$ and $w$ such that $||Z_i - w_j||$ is minimized.

## 2.1 PRE-TRAINING DNM

Since $||Z_i - w_j||$ depends on the parameters of the encoder $\theta$ and the weight vectors of the SOM $w$, we need to give an initial estimate to the parameters prior to jointly optimize them. We initialize $\theta$ with a denoising stacked auto-encoder (DeSAE). The DeSAE is trained using the greedy layer-wise algorithm, followed by fine-tuning all of the layers stacked together. In order to initialize the SOM weights at a given time $n$, we compute embedding $Z_i$ from $\mathcal{E}_\theta$ for all of the data points in the training set and update $w$ using:

$$w_j(n+1) = w_j(n) + \eta(n)H_{(u,j)}(n)(Z_i - w_j(n)) \tag{1}$$

where $\eta$ is a learning rate function of time and $H_{(u,j)}$ is the neighborhood function, which defines the topology around the best matching neuron $u$ with respect to a given neuron $j$ at time $n$. We chose a Gaussian kernel for the neighborhood function:

$$H_{(u,j)}(n) = exp\left(-\frac{||p_j - p_u||^2}{2\sigma^2(n)}\right), \quad \sigma(n) = \sigma_0 exp\left(-\frac{n}{\alpha}\right)$$

where $p_j$ is the position of neuron $j$ in the lattice and $\sigma(n)$ denotes time-decaying standard deviation with constant parameters $\sigma_0$ and $\alpha$.

## 2.2 JOINT PARAMETER OPTIMIZATION

In order to jointly optimize $\theta$ and $w$, we need to incorporate the error of the SOM, which is the distance between a given embedding and its best matching neuron, into the loss function of the auto-encoder. Therefore, the auto-encoder tries to generate a better embedding of the input data. We adopt the same idea from Xie et al. (2016) to use the Student's t-distribution to measure the similarity between $Z_i$ and $w_j$ and define a target distribution ($P$):

$$q_{ij} = \frac{\left(1 + ||Z_i - w_j||^2\right)^{-1}}{\sum_{j'}\left(1 + ||Z_i - w_{j'}||^2\right)^{-1}}, \quad p_{ij} = \frac{q_{ij}^2/\sum_i q_{ij}}{\sum_{j'} q_{ij'}^2/\sum_i q_{ij'}}$$

and minimize the KL divergence between $Q$ and $P$. The KL divergence minimization makes the probability of assigning $w_j$ to $Z_i$ closer to 0 and 1, hence stricter. We regularize the KL divergence by adding the reconstruction error term to the loss function. This regularization controls the parameters of the SOM and guarantees the convergence of the SOM. Therefore, the loss function of the auto-encoder is defined as:

$$\mathcal{L} = KLD(P||Q) + \gamma||X - \widehat{X}||_2^2 + \beta||\theta||_2 \tag{2}$$

Joint optimization of $\theta$ and $w$ is a two step algorithm. First, for all data points $X_i$ in a batch, we compute $Z_i$ using $\mathcal{E}_\theta$ and we update $w$ using Eq. 1. Later, we fix $w$ and minimize $\mathcal{L}$ with the stochastic gradient descent algorithm. We repeat this process for all of the data batches in the training set.

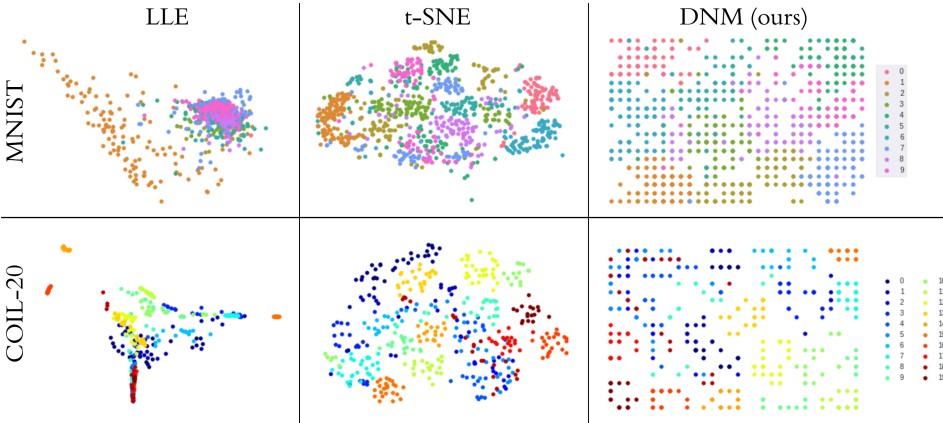

Figure 2: Comparison between LLE, t-SNE and DNM visualizations on MNIST and COIL-20 test data. DNM maps each class of the data to a particular region of the lattice, hence better separation of classes. The color maps are identical between methods for each data set.

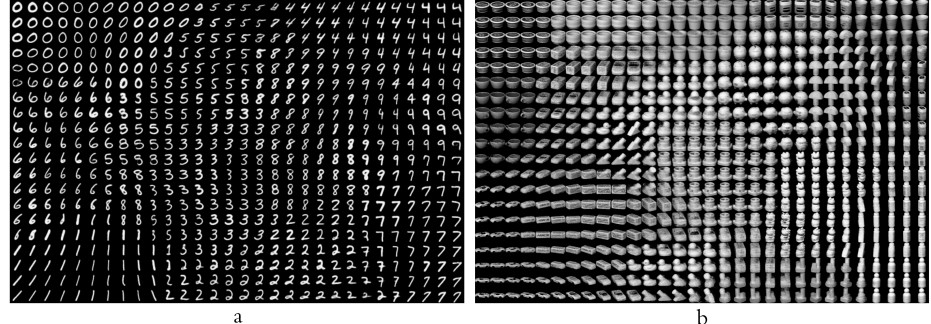

Figure 3: Back projection of the $20 \times 30$ SOM neurons $w$ from the lattice to the data space using $\mathcal{D}$ for MNIST (a) and COIL-20 (b). Despite its relatively simple architecture and small number of parameters, the DNM has successfully learned the style and pose variations of MNIST and COIL-20, respectively.

## 3 IMPLEMENTATION AND EXPERIMENTS

The auto-encoder filter dimensions are set to $(10 \times 5 \times 5)$-$(8 \times 5 \times 5)$-$(5 \times 5 \times 5)$-$dense100$ symmetrically for encoder and decoder for all of our experiments. We set the lattice size of the SOM to $20 \times 30$, which has 600 neurons in total and pre-trained the auto-encoder and the SOM for 500 and 10000 iterations, respectively with $\alpha = 2000$, $\sigma_0 = 10$ and $\eta(n) = 0.3 exp(-n/\alpha)$. Later, we trained the DNM using loss function 2 with $\gamma = 0.5$ and $\beta = 1e - 6$ for 500 iterations.

We evaluated the performance of the DNM on two widely used standard data sets, namely MNIST (LeCun et al. (2010)) and COIL-20 (Nene et al. (1996)). MNIST data set consists of 70000 hand-written digits of size $28 \times 28$ pixels (60000 train, 10000 test) and COIL-20 includes 20 objects, each of which has 72 images. We re-sized the original COIL-20 images to $64 \times 64$ and randomly selected 1000 images for training (70%). Figure 2 shows the results of our experiments on MNIST and COIL-20 test data. For the DNM model, the test data were excluded from training samples. We compared our method with t-SNE (Maaten & Hinton (2008)) and Locally Linear Embedding (LLE) (Roweis & Saul (2000)), which are two common techniques for high-dimensional data visualization. Note that the DNM has successfully separated each class of the data and mapped it to a particular location on the lattice. Figure 3 also shows the back-projection of the SOM from lattice to the data space for all of the neurons of the map. Note that the DNM model has successfully learned and captured the variations within each class of data and mapped them to their corresponding location in the lattice.

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
