# OpenReview forum: "Deep Neural Maps"
_ICLR.cc/2018/Workshop — Accept_

### Official Review · AnonReviewer1 · 2018-02-28
**Straight forward combination of a deep auto-encoder and a SOM.**

**Rating:** 6
**Confidence:** 3

**Review:**

The paper proposes a technique called Deep Neural Maps. It uses deep convolutional auto-encoder to compute encoding and decoding from high- to low-dimensional data. It also uses a self-organising map on the low-dimensional embedding to produce a 2D mapping. The technique jointly optimises the auto-encoder and the SOM.

The paper is well written and easy to follow. Visual analysis of high-dimensional data is an important problem. The novelty is limited though, since this is an ad hoc combination of an auto-encoder and a SOM.

Minor points:
The sentence "learns a projection of the data into a matrix of pair-wise similarities" is unclear.
Citation for DeSAE is missing.

---

### Official Review · AnonReviewer2 · 2018-03-09
**DAE + SOM**

**Rating:** 5
**Confidence:** 4

**Review:**

The paper combines two dimensionality reduction techniques: deep auto-encoders and self-organizing maps.
The goal is apparently to have a two-stage reduction of dimensionality.
First stage with the DAE to some target dimensionality, not necessarily very low.
Second stage to very low, such as two, for visualization purposes.

The is not very clear in its motivation.
In particular, the motivation to use a SOM is a bit obscure, as it is apparently justified by the need to have an OOSE (out of sample extension), to keep up with the parametric nature of SOM. Not sure the SOM is the best option. See e.g. parametric versions of t-SNE

There are a few typos (plurals in particular). There are notation clashes for the various sigmas. Van der Maaten is the last name of Laurens Van der Maaten, not Maaten only (see in the bib refs). Pay attention to ensure upper cases for acronyms in the Bib.

There is no qquantitative assessment of the experiments. In addition, the results for t-SNE are unusual for COIL; they should be better. Deep NNs still struggle to outperform t-SNE; the paper does not deal with this and it is not convincing in this respect.

---

### Official Review · AnonReviewer3 · 2018-03-12
**An interesting approach but lacks fair experimental comparisons**

**Rating:** 6
**Confidence:** 5

**Review:**

This paper proposes a method called Deep Neural Map (DNM) that uses
a combination of deep (denoising) autoencoder and self-organizing map
(SOM) for data embedding and visualization. The idea is interesting, but
the experimental results are not convincing.

Learning DNM consists of: (1) pre-training deep autoencoder and SOM
and (2) jointly optimizing the deep network parameters and SOM weight
vectors for sharper assignment of data points to map positions. The joint
optimization is performed in an alternating optimization fashion.

Compared to t-SNE, DNM has the advantage of embedding data into
a low-dimensional space with dimensionality much larger than two and
performing data visualization in a map simultaneously. The parametric
embedding facilities visualizing out-of-sample data points.Therefore, the
idea in this paper is interesting.

However, the visualization result of t-SNE is very poor, which is not
convincing at all. Detailed parameter settings and more fair comparisons
with existing methods such as t-SNE and pt-SNE (based on the same
deep denoising autoencoder) should be conducted.

---

### Decision · Program_Chairs · 2018-03-20
**ICLR 2018 Workshop Acceptance Decision**

**Decision:**

Accept

**Comment:**

Congratulations, your paper was accepted to the ICLR workshop.